# Assessing the effectiveness of a 3-month day-and-night home closed-loop control combined with pump suspend feature compared with sensor-augmented pump therapy in youths and adults with suboptimally controlled type 1 diabetes: a randomised parallel study protocol

Lia Bally,[1,2] Hood Thabit,[1,2,3] Martin Tauschmann,[1,4] Janet M Allen,[1,4] Sara Hartnell,[2] Malgorzata E Wilinska,[1,4] Jane Exall,[5] Viki Huegel,[6] Judy Sibayan,[6] Sarah Borgman,[6] Peiyao Cheng,[6] Maxine Blackburn,[7] Julia Lawton,[7] Daniela Elleri,[8] Lalantha Leelarathna,[3] Carlo L Acerini,[1,4] Fiona Campbell,[5] Viral N Shah,[9] Amy Criego,[10] Mark L Evans,[1,2] David B Dunger,[1,4] Craig Kollman,[6] Richard M Bergenstal,[10] Roman Hovorka[1,4]

► Prepublication history and additional material are available. To view these files please visit the journal online (http://dx.doi.org/10.1136/bmjopen-2017-016738).

For numbered affiliations see end of article.

**Correspondence to**
Dr. Roman Hovorka; rh347@cam.ac.uk

## ABSTRACT

**Introduction** Despite therapeutic advances, many individuals with type 1 diabetes are unable to achieve tight glycaemic target without increasing the risk of hypoglycaemia. The objective of this study is to determine the effectiveness of a 3-month day-and-night home closed-loop glucose control combined with a pump suspend feature, compared with sensor-augmented insulin pump therapy in youths and adults with suboptimally controlled type 1 diabetes.

**Methods and analysis** The study adopts an open-label, multi-centre, multi-national (UK and USA), randomised, single-period, parallel design and aims for 84 randomised patients. Participants are youths (6–21 years) or adults (>21 years) with type 1 diabetes treated with insulin pump therapy and suboptimal glycaemic control (glycated haemoglobin (HbA1c) ≥7.5% (58 mmol/mol) and ≤10% (86 mmol/mol)). Following a 4-week run-in period, eligible participants will be randomised to a 3-month use of automated closed-loop insulin delivery combined with pump suspend feature or to sensor-augmented insulin pump therapy. Analyses will be conducted on an intention-to-treat basis. The primary outcome is the time spent in the target glucose range from 3.9 to 10.0 mmol/L based on continuous glucose monitoring levels during the 3-month free-living phase. Secondary outcomes include HbA1c at 3 months, mean glucose, time spent below and above target; time with glucose levels <3.5 and <2.8 mmol/L; area under the curve when sensor glucose is <3.5 mmol/L, time with glucose levels >16.7 mmol/L, glucose variability; total, basal and bolus insulin dose and change in body weight. Participants' and their families' perception in terms of lifestyle change, daily diabetes management and fear of hypoglycaemia will be evaluated.

## Strengths and limitations of this study

► The study adopts a multi-centre, multi-national, randomised, parallel design
► The study includes youth and adult participants across a wide range of age groups and geographical locations
► Participants in both study arms have equal numbers of study visits
► The study excludes participants who are hypoglycaemia unaware, living alone and those with HbA1c below 7.5% and above 10%
► The study is open-label, and the comparator does not include the use of pump suspend feature

**Ethics and dissemination** Ethics/institutional review board approval has been obtained. Before screening, all participants/guardians will be provided with oral and written information about the trial. The study will be disseminated by peer-reviewed publications and conference presentations.

**Trial registration number** NCT02523131; Pre-results.

## INTRODUCTION

Type 1 diabetes mellitus (T1D) is caused by an immune-mediated destruction of insulin-producing pancreatic beta-cells resulting in insulin deficiency.[1] It commonly presents in childhood, but one-fourth of cases are diagnosed in adults. T1D accounts for 5%–10% of the total diabetes cases worldwide.[2] The

incidence of T1D is still increasing by around 3% every year, particularly among children.[2] Around 86 000 children develop T1D each year, and in 2015, the number of children with T1D exceeded half a million for the first time.[2]

Despite rapid advancements in insulin delivery and the on-going development of more physiological insulin preparations,[3] achieving optimal glycaemic control while avoiding hypoglycaemia remains a challenge for many people with T1D. The emergence of continuous glucose monitoring (CGM) over the last decade,[4 5] which enables users to view real-time interstitial glucose readings and receive alarms for impending hypoglycaemia or hyperglycaemia, facilitating appropriate changes in insulin therapy, is a major step towards improved diabetes monitoring. Studies have shown the clinical benefit of CGM in reducing HbA1c with regular use of the device.[6] Sensor-augmented pumps (SAPs) combine real-time CGM with insulin pump therapy, and studies have shown HbA1c reduction compared with multiple daily injection basal–bolus therapy.[7] Automatic suspension of insulin delivery by the pump when a predefined threshold is reached (low glucose suspend feature) has been demonstrated to significantly reduce the burden of hypoglycaemia.[8]

The development of a closed-loop system that combines glucose monitoring with computer-based algorithm-directed insulin delivery[9] may provide further improvements in glycaemic control while reducing hypoglycaemia[10 11] and represents an emerging treatment option for people with T1D.[12] Closed-loop differs from conventional pump therapy, characterised by preprogrammed basal delivery, through the use of a control algorithm that directs subcutaneous insulin delivery according to sensor glucose levels. People with T1D are known to have significant variability in daily insulin requirements,[13] which may be addressed by closed loop.

Closed-loop studies have progressed from fully supervised research facility settings[14 15] to remotely monitored transitional settings such as diabetes camp and hotel settings.[16 17] The use of closedloop during unsupervised free living conditions over extended periods represents the ultimate testing bed within the target environment (table 1). The longest randomised day-and-night home study to date has shown benefit in glycaemic control with concurrent reduction of the risk of hypoglycaemia.[10] On the basis of a recent non-randomised 3-month pivotal study,[12] the US Food and Drug Administration (FDA) approved the insulin-only closed-loop system for use in clinical practice.[18]

No study thus far has combined the pump suspend feature[8] with closed-loop insulin delivery. In previous closed-loop systems, preprogrammed basal insulin rates are delivered in the event of communication failure between control algorithm device and the insulin pump. In the present closed-loop system, in-built pump suspend feature is immune to such communication failures and will operate even if the control algorithm device is not in range, reducing the risk of prolonged hypoglycaemia. Thus, the addition of pump suspend feature to a closed-loop system further increases user safety by discontinuing insulin delivery at low sensor glucose levels.

We plan to assess the efficacy, safety, utility and acceptability of a 3-month day-and-night home closed-loop glucose control combined with pump suspend feature in youths and adults under free-living conditions compared with SAP therapy.

## METHODS AND ANALYSIS
### Overview

This is an open-label, multi-centre, multi-national, randomised, one-period, parallel study contrasting day-and-night automated closed-loop glucose control combined with pump suspend feature (threshold suspend) to therapy during free-living home setting (figure 1, protocol version 5.0 dated 28 February 2017). Study participants will include youths and adults with suboptimally controlled T1D treated with insulin pump therapy. Recruited participants will be randomly assigned to 3 months of study intervention. The study will aim to randomise 84 participants stratified at study site, as appropiate, according to the two age groups, which will be based on the age of minors in the USA (equal proportions of youths (6–21 years) and adults (22 years and older)). The University of Cambridge (UK) and Jaeb Center (USA) will be the coordinating centres. Clinical sites include:

1. Addenbrooke's Hospital, Cambridge, UK (adults and youths)
2. Royal Hospital for Sick Children, Edinburgh, UK (youths)
3. Leeds Teaching Hospital, Leeds, UK (youths)

**Table 1** Primary outcome of four randomised free-living unsupervised home day-and-night closed-loop studies

| Study | Duration of home closed loop (days) | Population | Time spent in target (%) | | p Value | Reference |
|---|---|---|---|---|---|---|
| | | | Closed loop | Control | | |
| Dan04 phase 1 | 7 | Adolescents | 72 | 53 | <0.001 | Tauschmann et al[24] |
| Dan04 phase 2 | 21 | Adolescents | 67 | 48 | <0.001 | Tauschmann et al[25] |
| AP@home02 | 7 | Adults | 75 | 63 | 0.006 | Leelarathna et al[23] |
| AP@home04 phase 1 | 84 | Adults | 68 | 57 | <0.001 | Thabit et al[10] |
| AP@home04 phase 2 | 28 | Adults | 76 | 66 | <0.001 | Bally et al[26] |

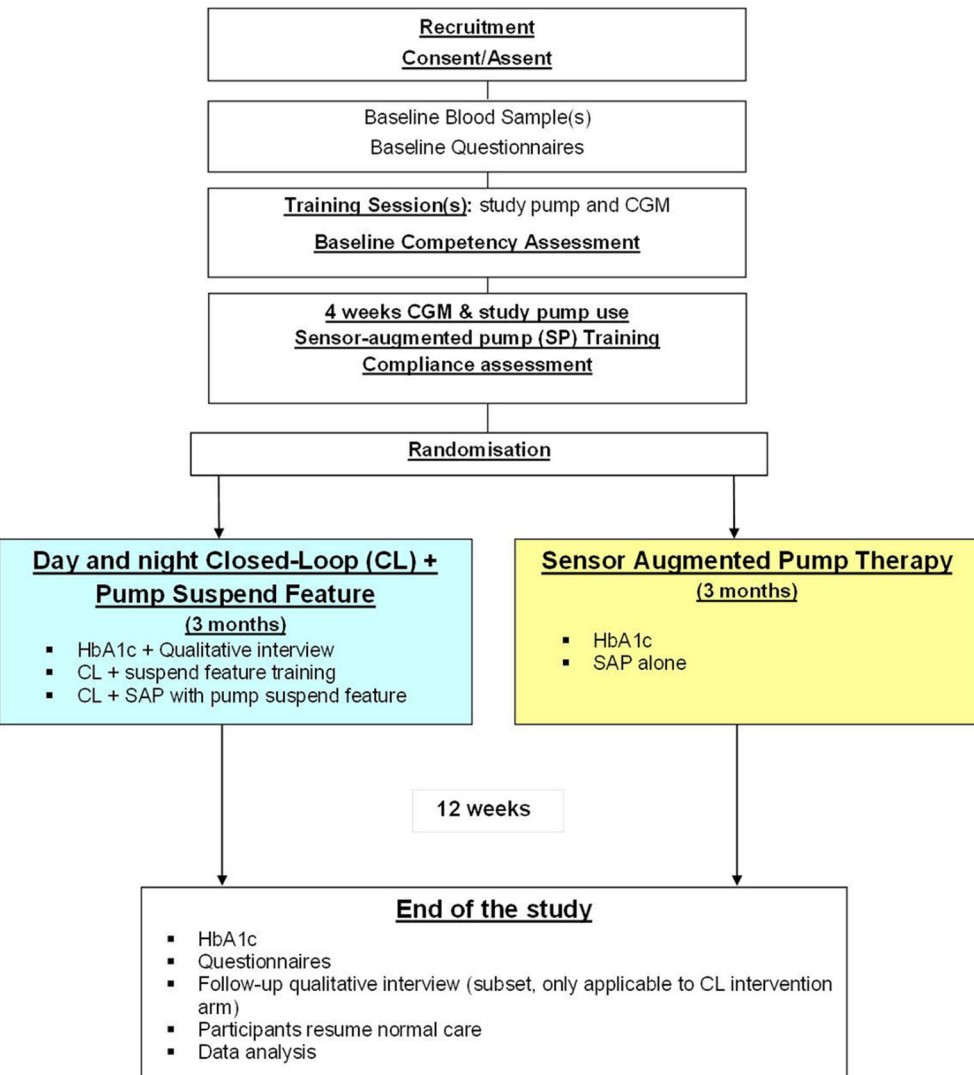

**Figure 1** Study flow chart. CGM, continuous glucose monitoring; HbA1c, glycated haemoglobin.

4. Manchester Royal Infirmary, Manchester, UK (adults)
5. International Diabetes Center at Park Nicollet, Minneapolis, USA (adults and youths)
6. Barbara Davis Center for Childhood Diabetes, Aurora, USA (adults)

Qualitative interviews will be carried out by the University of Edinburgh, UK. Written informed consent/assent will be obtained from all participants and guardians, as appropriate, before any study-related activities.

### Inclusion criteria

▶ At least 6 years or older (with equal proportion of youths (6–21 years) and adults (22 years and older))
▶ T1D as defined by WHO for at least 1 year or confirmed C peptide negative
▶ An insulin pump user for at least 3 months, with good knowledge of insulin self-adjustment as judged by the investigator
▶ Treated with one of the U-100 rapid-acting insulin analogues only (insulin aspart, lispro but not glulisine)

▶ Willing to perform regular capillary blood glucose monitoring, with at least four blood glucose measurements taken every day
▶ Screening HbA1c ≥7.5% (58.5 mmol/mol) and ≤10% (86 mmol/mol) based on analysis from local laboratory or equivalent
▶ Literate in English
▶ Willing to wear CGM
▶ Willing to wear closed-loop system at home
▶ Willing to follow study-specific instructions
▶ Willing to upload pump and CGM data at regular intervals
▶ Willing to restrict alcohol consumption to ≤2 units per day throughout the study period
▶ Female participants of child-bearing age should be on effective contraception and must have a negative urine-HCG pregnancy test at screening
▶ Living with someone who is trained to administer intramuscular glucagon and is able to seek emergency assistance
▶ Access to Wi-Fi at home.

## Exclusion criteria

- Non-T1D including those secondary to chronic disease
- Using real-time CGM on regular basis in preceding 3 months
- Any other physical or psychological disease likely to interfere with the normal conduct of the study and interpretation of the study results as judged by the investigator
- Untreated coeliac disease, adrenal insufficiency or hypothyroidism
- Current treatment with drugs known to interfere with glucose metabolism, for example, systemic corticosteroids, non-selective beta-blockers and monoamine oxidase inhibitors, among others
- Known or suspected allergy to insulin
- Clinically significant nephropathy (estimated glomerular filtration rate <45 mL/min) or on dialysis, neuropathy or active retinopathy (defined as the presence of maculopathy or proliferative changes) as judged by the investigator
- Adults: one or more episodes of severe hypoglycaemia as defined by American Diabetes Association[19] in preceding 6 months; youths: one or more episodes of severe hypoglycaemia during the previous 6 months (adults and adolescents: severe hypoglycaemia is defined as an event requiring assistance of another person to actively administer carbohydrates or glucagon or take other corrective actions including episodes of hypoglycaemia severe enough to cause unconsciousness, seizures or attendance at hospital; children: severe hypoglycaemia is defined as an event associated with a seizure or loss of consciousness)
- Random C peptide >100 pmol/L with concomitant blood glucose >4 mmol/L (72 mg/dL)
- Regular use of acetaminophen
- Lack of reliable telephone facility for contact
- Total daily insulin dose ≥2 IU/kg/day
- Total daily insulin dose <15 IU/day
- Pregnancy, planned pregnancy or breast feeding
- Severe visual impairment
- Severe hearing impairment
- Significantly reduced hypoglycaemia awareness in participants aged 18 years and older defined by Gold score of >4
- Using implanted internal pace-maker
- Medically documented allergy towards the adhesive (glue) of plasters or unable to tolerate tape adhesive in the area of sensor placement
- Serious skin diseases (eg, psoriasis vulgaris, bacterial skin diseases) located at places of the body, which potentially are possible to be used for the localisation of glucose sensor
- Currently abusing alcohol, illicit drugs or prescription drugs
- Using pramlintide (symlin) or other commonly used hypoglycaemic agents including sulphonylureas, biguanides, DPP4 inhibitors, GLP-1 agonists, SGLT-2 inhibitors at time of screening
- Elective surgery planned that requires general anaesthesia during the course of the study
- Shift worker with working hours between 22:00 and 08:00
- Sickle cell disease or haemoglobinopathy
- Red blood cell transfusion or erythropoietin within 3 months prior to the time of screening or plans to receive red blood cell transfusion or erythropoiesstin during the course of study participation
- Diagnosed with current eating disorder such as anorexia or bulimia
- Plans to use significant quantity of herbal preparations (use of over-the-counter herbal preparation for 30 consecutive days or longer period during the study) or significant quantity of vitamin supplements (four times the recommended daily allowance used for 30 consecutive days or longer period during the study), which may affect glucose metabolism and/or blood glucose levels during the course of their participation in the study.

## Study schedule

The study will consist of up to 11 in-clinic visits and 6 preplanned telephone/email contacts. Overview and key activities of the study visits are shown in tables 2 and 3, with both intervention periods lasting 3 months.

## Study training

Following written informed consent, participants will be trained on the use of study insulin pump (modified Medtronic 640G pump, Medtronic MiniMed, Northridge, California, USA) and CGM sensor (Enlite 3, Medtronic). For minors, parents/guardians will provide written informed consent, and written assent will be obtained from minors. Training will be performed in the presence of parents/guardians. The study insulin pump will be preprogrammed using participants' usual basal settings, carbohydrate-to-insulin ratio and insulin sensitivity factor. Competency on use of study devices with the participant (cosignature of guardian will be required in US participants) will be formally assessed.

## Run-in period

There will be a minimum 4-week run-in period. Participants will be contacted once weekly by phone or email for troubleshooting purposes. At the end of the run-in period, study devices will be downloaded to assess adherence during the last 14 days (participants will be required to demonstrate at least 12 days' use of CGM and use of at least 75% of bolus calculator for meal boluses for randomisation). Participants need to perform weekly downloads of study devices.

## Randomisation

Eligible subjects will be randomised using central randomisation software to the use of day-and-night closed-loop

**Table 2** Schedule of study visits/phone contacts when the participant is randomised to day-and-night closed -loop combined with pump suspend feature (intervention group)

| | Visit/contact | Description | Start relative to previous/next visit/activity | Duration, hours |
|---|---|---|---|---|
| Training and run-in (4 weeks) | Visit 1 | Recruitment visit: consent, HbA1c, screening bloods and questionnaires, urine pregnancy test | - | 1–4 |
| | Visit 2 | Insulin pump training and the initiation of study pump, competency assessment | Within 1–3 weeks of visit 1 | 3–4 |
| | Visit 3 | CGM training, initiation of CGM, weekly contact via phone/email, competency assessment | Within 3–7 days of visit 2 | 2–3 |
| | Visit 4* | Review pump and CGM data, optimisation of treatment, compliance assessment, randomisation | After 4 weeks of visit 3 | <1 |
| | Contact | Qualitative interview (with a subset of participants/family members) | After randomisation but before visit 5 | <1 |
| CL + LGS intervention (3 months) | Visit 5 | CL initiation at clinic/home: urine pregnancy test, CL and suspend feature training, competency assessment, HbA1c | Within 1 week of visit 4 | 2–6 |
| | Visit 6* | Review use of study devices | After 24–48 hours of visit 5 | <1 |
| | Visit 7† | Review use of study devices | After 1 week of visit 5 | <1 |
| | Visit 8* | Review pump and CGM data | After 1 week of Visit 7 | <1 |
| | Visit 9* | End of first month: review pump and CGM data | After 2 weeks of visit 8 | <1 |
| | Visit 10* | End of second month: review pump and CGM data | After 4 weeks of visit 9 | <1 |
| | Visit 11 | End of closed-loop treatment arm (3 months): HbA1c, complete questionnaires and follow-up qualitative interviews | After 4 weeks of visit 10 | 1–3 |

*Could be done via phone/e-mail in the UK. Follow-up by phone is mandatory in the USA only.
†Could be done via phone/e-mail in the UK. In-person visit mandatory in USA only.
CGM, continuous glucose monitoring; CL, closed loop; HbA1c, glycated haemoglobin; LGS, low glucose suspend.

combined with pump suspend feature (threshold suspend) or to sensor-augmented insulin pump therapy. The randomisation will be stratified at each centre by the two age groups (equal proportion of youth (6–21 years) and adults (22 years and older)) and HbA1c groups (equal proportion of below and above 8.5% (69 mmol/mol)).

A subset of participants randomised to closed-loop will be invited to take part in an interview study led by an independent team of qualitative researchers at the University of Edinburgh. Participants will be interviewed before and/or at the end of the closed-loop study period.

### Home treatment period
All participants will be admitted to a clinical facility at the start of each treatment period and have blood tests performed for HbA1c. Females of child-bearing age will have urine pregnancy test on arrival.

### Day-and-night closed-loop combined with pump suspend feature
Participants will be trained on connection and disconnection of the closed-loop system and switching between closed-loop and usual pump therapy. Participants will be instructed on the use of low-glucose suspend functionality and settings. During closed-loop period, meal bolus will be delivered by the insulin pump based on carbohydrate estimation. Specific instructions during closed-loop related to exercise management, sick day rules, hypoglycaemia and hyperglycaemia management and technical troubleshooting will also be reviewed with the participants by the study team. Competency on the safe and effective use of the closed-loop system will be assessed by the study team; only those who demonstrate the same will be allowed to continue to the home study phase.

### Sensor-augmented pump (SAP) therapy
Participants randomised to SAP therapy will receive training on the effective use of real-time CGM for optimisation of insulin therapy. During SAP therapy, participants

**Table 3** Schedule of study visits/phone contacts when the participant is randomised to sensor-augmented pump

| | Visit/contact | Description | Start relative to previous/ next visit/activity | Duration, hours |
|---|---|---|---|---|
| Training and run-in (4 weeks) | Visit 1 | Recruitment visit: consent HbA1c, screening bloods and questionnaires, urine pregnancy test | – | 1–4 |
| | Visit 2 | Insulin pump training and the initiation study pump, competency assessment | Within 1–3 weeks of visit 1 | 3–4 |
| | Visit 3 | CGM training, initiation of CGM, weekly contact via phone/email, competency assessment | Within 3–7 days of visit 2 | 2–3 |
| | Visit 4* | Review pump and CGM data, optimisation of treatment, compliance assessment, randomisation | After 4 weeks of visit 3 | <1 |
| SAP intervention (3 months) | Visit 5 | SAP initiation at clinic/home: urine pregnancy test, SAP training, competency assessment, HbA1c | Within 1 week of visit 4 | 2–6 |
| | Visit 6* | Review use of study devices | After 24–48 hours of visit 5 | <1 |
| | Visit 7† | Review use of study devices | After 1 week of visit 5 | <1 |
| | Visit 8* | Review pump and CGM data | After 1 week of visit 7 | <1 |
| | Visit 9* | End of first month: review pump and CGM data | After 2 weeks of visit 8 | <1 |
| | Visit 10* | End of second month: review pump and CGM data | After 4 weeks of visit 9 | <1 |
| | Visit 11 | End of SAP treatment arm (3 months): HbA1c, complete questionnaires | After 4 weeks of visit 10 | 1–3 |

*Could be done via phone/e-mail in the UK. Follow-up by phone is mandatory in the USA only.
†Could be done via phone/e-mail in the UK. In-person visit mandatory in the USA only.
CGM, continuous glucose monitoring; SAP, sensor-augmented pump; HbA1c, glycated haemoglobin.

will enter information related to carbohydrate content of the meals and premeal capillary glucose values into the bolus wizard of the insulin pump. Participants will be instructed not to activate threshold suspend (low-glucose or predictive low-glucose suspend features). Hypoglycaemia and hyperglycaemia alarms can be activated and personalised as per participants' requirements.

### Contact during 3-month home study period

Participants will have identical planned contact visits with the study team during the two study periods. The first planned contact will occur 24–48 hours once study period has begun. During the first 2 weeks of the study period, participants will be contacted weekly. Thereafter, participants will be contacted monthly. Participants will be free to optimise their treatment independently.

### Devices download

Participants will be instructed to download insulin delivery and glucose data weekly from the insulin pump and blood glucose metre using the Medtronic Carelink Clinical download facility.

### Closed-loop system and low glucose suspend feature

The FlorenceM closed-loop system (figure 2, University of Cambridge, Cambridge, UK) comprises a model predictive control algorithm residing on an Android smartphone that communicates wirelessly with the insulin pump using a proprietary enclosure.

Every 10 min, the system calculates a new temporary basal insulin infusion rate, which is automatically sent to

the study insulin pump. Calculations use a compartment model of glucose kinetics describing the effect of rapid-acting insulin analogues and the carbohydrate content of meals on glucose levels.[20] The control algorithm will be initialised using preprogrammed basal insulin delivery downloaded from the study pump. In addition, information about participants' weight and total daily insulin dose will be entered at setup. During closed-loop operation, the algorithm adapts itself to a particular participant. The treat-to-target control algorithm aims to achieve a default

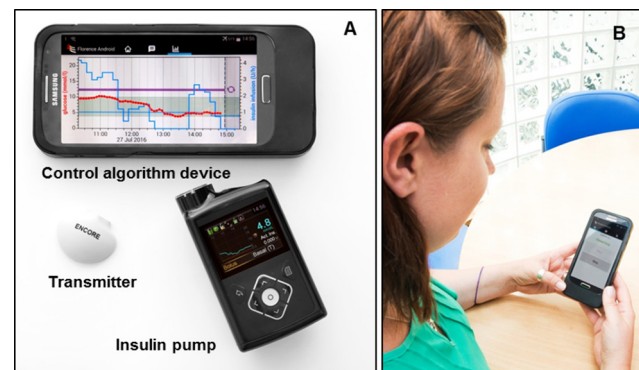

**Figure 2** Closed-loop system prototype. (A) Components of the FlorenceM closed-loop system consist of a continuous glucose monitoring (CGM) transmitter with Enlite 3 sensor, an insulin pump (modified 640G pump) integrated with the CGM receiver and a mobile phone running the control algorithm. (B) A photo of a participant (obtained with consent) using the closed-loop system.

glucose level of around 5.8 mmol/L and adjusts the actual level depending on fasting versus postprandial status and the accuracy of model-based glucose predictions. Sensor glucose, insulin delivery, carbohydrate content and other information can be visualised by the participants on the smartphone device.

No remote monitoring is instigated. For data collection purposes, the control algorithm device uploads data on a server when the system is charging and connected to Wi-Fi.

The pump comprises a CGM receiver and provides hypoglycaemia and hyperglycaemia alarms. Hypoglycaemia and hyperglycaemia alarms can be activated and personalised as per participants' requirements. The insulin pump provides standard insulin pump alarms, and the smartphone alerts the user about aspects related to closed-loop operations (closed-loop started, stopped or terminated).

### Safety precautions during closed-loop
Participants will be trained to perform calibration checks before breakfast and evening meals. If sensor glucose is above finger-stick glucose by more than 3.0 mmol/L, the glucose sensor will be recalibrated. These instructions resulted from an in-silico simulation of hypoglycaemia and hyperglycaemia risk using the validated Cambridge simulator.[21]

If sensor glucose becomes unavailable, preprogrammed insulin delivery will be automatically started within 30 min. This limits the risk of insulin underdelivery and overdelivery. Safety rules limit maximum insulin infusion and suspend insulin delivery at sensor glucose at or less than 4.3 mmol/L or when sensor glucose is rapidly decreasing.

In the event of closed-loop becoming unavailable due to loss of sensor glucose input or other system failures, insulin delivery will be suspended by the low glucose suspend feature set on the insulin pump, provided sensor glucose is available. Resumption of insulin delivery will be in accordance of the low glucose suspend feature implemented on 640G pump.

### Participant withdrawal criteria
The following prerandomisation withdrawal criteria will apply:
1. Participant is unable to demonstrate safe use of study insulin pump and/or CGM during run-in period as judged by the investigator
2. Participant fails to demonstrate compliance with study insulin pump and/or CGM during run-in period.

The following prerandomisation and postrandomisation withdrawal criteria will apply:
1. Participants may terminate participation in the study at any time without necessarily giving a reason and without any personal disadvantage
2. Significant protocol violation or non-compliance
3. Any severe hypoglycaemia event related to use of the closed-loop system

4. Three severe hypoglycaemia events unrelated to the use of the closed-loop system
5. Diabetic ketoacidosis (DKA) unrelated to infusion site failure and related to the use of the closed-loop system
6. Decision by the investigator or the sponsor that termination is in the participant's best medical interest
7. Participant becomes pregnant during the study period
8. Allergic reaction to insulin
9. Allergic reaction to adhesive surface of infusion set or CGM sensor
10. If patient continues to use pump suspend feature in the control group despite advice to the contrary.

Withdrawn participants due to reasons 4–12 will be invited to provide blood sample at the end of the planned study intervention for the assessment of HbA1c.

### Psychosocial methods
A mixed-methods psycho-social evaluation using in-depth interviews and validated questionnaires will be conducted to determine the utility of the device in terms of acceptability of intervention, quality of life, participants' perception of impact on lifestyle and diabetes self-management.

Participants will be invited to complete the PedsQoL questionnaire at screening and at the end of the study intervention. In addition, a feedback questionnaire on closed-loop-specific experience will be distributed to participants/guardians who had been randomised to the closed-loop intervention arm.

A subset of participants (individuals aged 16+ years, individuals aged 13–15 years and their parent/guardian(s), parent/guardian(s) of participants aged 12 years and under) will be invited to take part in an interview at baseline (postrandomisation) to enable their historical diabetes management practices, everyday work/school and family lives and their initial understandings and expectations of using closed-loop technology to be captured and explored in depth. The same participants will be followed-up at the end of the study intervention to determine whether the use of the closed loop-matched expectations, whether any difficulties occurred, the benefits and downsides of using closed loop to support diabetes self-management for the duration of the trial and their views about how the technology might be improved.

A subset of partners/family members will be interviewed at 3 months to capture the benefits of using a closed-loop system from their perspectives, how this technology has impacted on their own lives and on their role in supporting diabetes management practices.

Recruitment to the interview study will continue until data saturation is achieved; that is, until no new findings are identified in new data collected. It is estimated that around 20–25 adults and youths with diabetes and 20 members will be interviewed. If necessary, participants will be purposely sampled to allow for diversity in the

final sample in terms of age, gender, occupation/education and geographical location. Interviews will be digitally recorded with consent.

## HbA1c samples

Blood samples for the measurement of HbA1c levels will be taken at three different time points. At screening, HbA1c will be measured locally. HbA1c at the beginning and end of study intervention will be taken and measured at a central laboratory (University of Minnesota, Minneapolis, USA) using an International Federation of Clinical Chemistry and Laboratory Medicine aligned method.

## Statistical analysis

All analysis will be conducted on an intention-to-treat basis. Data from all randomised participants with or without protocol violation including dropouts and withdrawals will be included in the analysis. Data will not be truncated due to protocol deviations.

### Primary endpoint

The primary analysis will evaluate between group difference in the time (midnight to midnight) spent in the target glucose range from 3.9 to 10.0 mmol/L (70–180 mg/dL) based on sensor glucose levels during the 12-week free-living phase.

A 5% significance level will be used to declare statistical significance for the primary outcome comparison.

Mean (SD) for % time spent in the target range will be tabulated by treatment group. A linear model will be used to compare the difference between the two intervention arms, while adjusting for baseline % time spent in the target glucose range and random site effect. A 95% CI will be reported for the difference between the treatment groups based on the linear model. Normality of the residuals will be assessed. If the residuals have highly skewed distribution, then ranked normal score transformation of outcome data will be applied in the regression model. However, previous experience suggests that % time in target glucose range will follow an approximately normal distribution. A detailed analysis plan will be provided separately.

### Secondary endpoints

The following outcomes on the two treatment arms will be compared:
► HbA1c at 12 weeks
► Time spent below target glucose (3.9 mmol/L) (70 mg/dL)
► Time spent above target glucose (10.0 mmol/L) (180 mg/dL)
► Average, SD and coefficient of variation of glucose levels
► Time with glucose levels <3.5 mmol/L (63 mg/dL) and <2.8 mmol/L (50 mg/dL)
► Time with glucose levels in the significant hyperglycaemia (glucose levels >16.7 mmol/L) (300 mg/dL)

► Total, basal and bolus insulin dose (units/kg of body weight)
► Area under the curve of glucose below 3.5 mmol/L (63 mg/dL)
► Number of pump suspend events (applicable to intervention arm)
► Change in body weight from screening to end of study.

Glycaemic metrics will be based on sensor glucose levels collected during the 12-week intervention period. Similar linear models as the primary outcome will be used to compare the between-treatment difference. For those metrics that have highly skewed distribution, a ranked normal score transformation of outcome data will be applied in the regression model. p Value <0.05 will be used to declare statistical significance for selected secondary outcomes (HbA1c, glucose CV, % time below 3.9 mmol/L, % time above 10.0 mmol/L, total daily insulin and change in body weight). For other outcomes, to reduce the inflation of type I error caused by multiple comparisons, p value <0.01 will be used to define statistical significance.

A subset of sensor glucose and insulin metrics will also be tabulated separately for daytime period (08:00–00:00) and night-time period (00:00–08:00).
► Percent time with glucose in target range of 3.9–10.0 mmol/L (70–180 mg/dL)
► Mean glucose
► Glucose variability as measured by SD
► Percent time with glucose level <3.5 mmol/L (63 mg/dL)
► Amount of delivered insulin.

Trends in sensor glucose data collected within intervention arms will be evaluated on a 4-weekly basis.

## Safety analysis

The following safety outcomes will be tabulated by treatment group:
► Number of subjects with any DKA events
► Number of episodes of DKA events per subject and incidence rate per 100 person-years
► Number of subjects with any severe hypoglycaemia events
► Number of episodes of severe hypoglycaemia events per subject and incidence rate per 100 person-years
► Number of subjects with any severe hyperglycaemia events as defined by fingerprick glucose >16.7 mmol/L (>300 mg/dL) and plasma ketones >0.6 mmol/L
► Number of episodes of severe hyperglycaemia events per subject and incidence rate per 100-person years.

Above safety data will be tabulated for all subjects in the two intervention periods, including dropouts and withdrawals, irrespective of whether sensor glucose data are available and irrespective of whether closed loop was operational. All adverse events will be listed for the entire study duration.

 Bally L, et al. BMJ Open 2017;7:e016738. doi:10.1136/bmjopen-2017-016738

If there are enough observed events to allow formal statistical modelling for above safety outcomes, the following analyses will be conducted. The event rates will be compared using a Poisson regression model adjusting for random site effect. If outliers exist, a robust Poisson regression model will be used instead. Binary variables will be compared using a logistic regression model adjusting for random site effect. Models involving severe hypoglycaemia events will also adjust for severe hypoglycaemia in the previous 6 months before enrolment.

## Utility evaluation

The amount of CGM use will be calculated for both intervention arms, and the amount of closed-loop system used will be calculated for the closed-loop treatment arm only over the 12-week intervention and by 4-weekly period. The difference in CGM use between treatment groups will be compared using similar linear model as described for primary outcome.

## Subgroup analysis

No subgroups were considered during the power calculations. Interpretation of any subgroup analyses will depend on whether the overall analysis demonstrates a significant treatment group difference. In the absence of such an overall difference and if performed, the following subgroup analyses will be interpreted with caution: (1) age (6–12 years, 13–21 years, 22 years and older), (2) gender, (3) race–ethnicity, (4) clinical centre and (5) baseline HbA1c (<8.5% and ≥8.5%).

## Psychosocial evaluation

Questionnaires for diabetes-related quality-of-life assessment (PedsQL v3.2) will be collected at screening and at the end of the study intervention. For subjects ≥18 years of age, only answers from the participants themselves will be collected; for subject s≤17 years of age, answers will be collected from both parent version and child version.

At each assessment time point, mean±SD score for each dimension and the total score will be tabulated by intervention arm for both parent version and participant version. The between-group difference of each score at end of study will be assessed using a similar linear model as described previously by adjusting for corresponding score at baseline.

Interview data will be analysed thematically and longitudinally using the method of constant comparison.[22] To maximise rigour, at least two experienced qualitative researchers will be involved in the analysis. NVivo9, a qualitative software package, will be used to facilitate data coding/retrieval.

## Per-protocol analysis

Per-protocol analysis will be conducted to compare the primary endpoint limited to participants who used closed loop for at least 80% of the of the 3-month study period (with CGM data are available for at least 50% of the study period). In the SAP group, per-protocol analysis will be conducted on those whose CGM data are available for at least 50% of study period.

## Interim analysis

No interim analysis will be performed.

## Power calculation

On the basis of our previous day-and-night closed-loop studies[10 23] and a conservative estimate of 10%points improvement in time when glucose is in target glucose range with an SD of 14.5% points, 76 subjects are required to achieve 85% power and an alpha level of 0.05 (two tailed). A total of 84 are planned to be randomised to allow for dropouts. It is expected that 95 or more subjects will be recruited; recruitment will continue until it is anticipated that a sufficient number have been enrolled to meet the randomisation goal. Randomisation target may be exceeded since all of the subjects who have initiated the run-in phase will be permitted to continue into the intervention.

# STUDY MANAGEMENT
## Data Safety Monitoring Board

An independent Data Safety Monitoring Board (DSMB) will comprise a chairperson and two experts. DSMB will be informed of all serious adverse events and any unanticipated device effects/events that occur during the study. DSMB will review compiled adverse event data and periodic intervals. DSMB will report to the study management committee any safety concerns and recommendations for suspension or early termination of the investigation. Composition of DSMB is shown in online supplementary appendix file 1 .

## Study sponsors

In the UK, the study sponsor is the Cambridge University Hospitals NHS Foundation Trust, jointly with the University of Cambridge. Study sponsor in the USA is the Jaeb Center for Health Research.

## Study management committee

The study management committee consisting of the chief investigator, study coordinators and study data manager will confer at least quarterly to discuss the operational aspects of the study. The principal clinical investigators may also participate. Composition of the study management committee is shown in online supplementary appendix.

## Study monitoring

The study coordinators will ensure that the study is conducted in accordance with Good Clinical Practice standards through site monitoring visits. A monitoring plan will be written and agreed prior to randomisation.

## Data management

Confidentiality of participant data shall be observed at all times during the study. Personal details for each participant taking part in the research study and linking them

to a unique identification number will be held locally on a study screen log in the Trial Master File at each of the investigation centres. These details will not be revealed at any other stage during the study, and all results will remain anonymous. Study identification number and bar codes will be used on the electronic case report forms and all the blood samples that are collected throughout the study.

Electronic data will be stored on password-protected computers. All paper records will be kept in locked filing cabinets, in a secure office at each of the investigation centres. Only members of the research team and collaborating institutions will have password access to the anonymised electronic data. Only members of the research teams will have access to the filing cabinet. Paper copies of the data will be stored for 15 years.

Direct access to the source data will be provided for monitoring, audits, research ethics committee (REC)/ institutional review board (IRB) review and regulatory authority inspections during and after the study. The fully anonymised data may be shared with third parties (EU or non-EU based) for the purposes of advancing management and treatment of diabetes.

### Indemnity

Indemnity for any harm rising on the conduct of research will be provided according to local arrangements in respective countries.

1. UK: any liability arising from study design will be covered by the clinical trial insurance policy organised by the University of Cambridge. National Health Service indemnity cover will apply for any claims arising from management and conduct of research.
2. USA: any liability arising from study design will be under the responsibility of the participants or their insurance company.

### ETHICS AND DISSEMINATION

The study has received approval from independent REC and IRB in the UK and the USA. The study has undergone a review by regulatory authorities in the UK (Medicines and Healthcare products Regulatory Agency) and in the USA (FDA). All participants will be provided with oral and written information about the trial and procedures involved in the study before obtaining written informed consent.

Study operating procedures for monitoring and reporting of all adverse events and adverse device effects will be in place including serious adverse events, serious adverse device effects and specific adverse events such as severe hypoglycaemia and significant hyperglycaemia with ketosis. DSMB will be informed of all serious adverse events and any unanticipated adverse device/method effects that occur during the study and will review compiled adverse events data at periodic intervals.

Any substantial amendments to the protocol and other documents shall be notified to, and approved by,

the independent REC and IRB (UK, Cambridge East Research Ethics Committee (#15/EE/0324); USA, Jaeb Center for Health Research Institutional Review Board certified by the Office for Human Research Protections (FWA #00000024)) and the regulatory authorities, prior to implementation as per nationally agreed guidelines.

Screening and recruitment commenced in June 2016, and the study is expected to be completed by the end of 2017. The study results will be disseminated by peer-reviewed publications and conference presentations.

**Author affiliations**
[1]Wellcome Trust-MRC Institute of Metabolic Science, University of Cambridge, Cambridge, UK
[2]Department of Diabetes & Endocrinology, Cambridge University Hospitals NHS Foundation Trust, Cambridge, UK
[3]Manchester Diabetes Centre, Central Manchester University Hospitals NHS Foundation Trust, Manchester Academic Health Science Centre, Manchester, UK
[4]Department of Paediatrics, University of Cambridge, Cambridge, UK
[5]Leeds Children's Hospital, Leeds, UK
[6]Jaeb Center for Health Research, Tampa, Florida, USA
[7]Centre for Population Health Sciences, University of Edinburgh, Edinburgh, UK
[8]Royal Hospital for Sick Children, Edinburgh, UK
[9]Barbara Davis Center for Diabetes, University of Colorado Denver, Aurora, Colorado, USA
[10]International Diabetes Center at Park Nicollet, St Louis Park, Minnesota, USA

**Acknowledgements** Jasdip Mangat supported development and validation of closed-loop system. Josephine Hayes (Institute of Metabolic Science, University of Cambridge) provided administrative support.

**Contributors** RH, HT, MT, DBD, MLE, LL, AC, RMB and VNS codesigned the study. JL was responsible designing the interview study. RH designed and implemented the glucose controller. LB, HT and RH wrote the manuscript. All authors critically reviewed the report. No writing assistance was provided.

**Funding** JDRF, National Institute for Health Research Cambridge Biomedical Research Centre, Wellcome Strategic Award (100574/Z/12/Z).

**Competing interests** SH serves as a consultant for Novo-Nordisk and for the ONSET group and reports having received speaker/training honoraria from Medtronic. MLE reports having received speaker honoraria from Abbott Diabetes Care, Novo Nordisk and Animas and serving on advisory panels for Novo Nordisk, Abbott Diabetes Care, Medtronic, Roche and Cellnovo. RH reports having received speaker honoraria from Eli Lilly, Novo Nordisk and Astra Zeneca; serving on advisory panel for Eli Lilly and Novo Nordisk; receiving license fees from BBraun and Medtronic and having served as a consultant to BBraun. MEW has received license fees from Becton Dickinson and has served as a consultant to Beckton Dickinson. MT reports having received speaker honoraria from Novo Nordisk and Medtronic. RH and MEW report patent patents and patent applications. LL reports having received speaker honoraria from Minimed Medtronic, Animas, Sanofi and Novo Nordisk and serving on advisory panel for Animas Minimed Medtronic and Novo Nordisk. VNS had received speaking fees from the Dexcom Inc. VNS employer received research funding from T1D Exchange and Sanofi. LB, HT, JMA, JE, VH, JS, SB, PC, MB, JL, DE, CLA, FC, AC, MLE, DBD, CK and RMB declare no competing financial interests exist.

**Patient consent** Obtained.

**Ethics approval** East of England-Cambridge East Research Ethics Committee.

**Provenance and peer review** Not commissioned; externally peer reviewed.

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
