## [Reviewer comments · BMJ Open]

ARTICLE DETAILS

TITLE (PROVISIONAL)	Assessing the effectiveness of 3 months day-and-night home closed-loop control combined with pump suspend feature compared to sensor augmented pump therapy in youths and adults with sub-optimally controlled type 1 diabetes: a randomised parallel study protocol
AUTHORS	Bally, Lia; Thabit, Hood; Tauschmann, Martin; Allen, Janet; Hartnell, Sara; Wilinska, Malgorzata; Exall, Jane; Huegel, Viki; Sibayan, Judy; Borgman, Sarah; Cheng, Peiyao; Blackburn, Maxine; Lawton, Julia; Elleri, Daniela; Leelarathna, Lalantha; Acerini, Carlo; Campbell, Fiona; Shah, Viral; Criego, Amy; Evans, Mark; Dunger, David; Kollman, Craig; Bergenstal, Richard; Hovorka, Roman

VERSION 1 - REVIEW

REVIEWER	Pierre Y. Benhamou University Grenoble Alpes, France member of the scientific board of the french Artificial Pancreas Diabeloop consortium
REVIEW RETURNED	26-Mar-2017

GENERAL COMMENTS	The primary objective of this multicenter and international trial is to determine the effectiveness of a closed-loop glucose control system in youths and adults with sub-optimally controlled type 1 diabetes. The strength of the study is to propose a randomised, parallel design trial, as opposed to traditional crossover designs that have been reported so far in this field. The duration of this home study (3 months) will yield clinically relevant informations. The metabolic indicators are clearly informative, and will include HbA1c. A psychosocial evaluation is included and should clarify the question of the acceptability of such a device. The feasibility and overall relevance of this study do not raise any doubt. Overall this trial is very important for the field and should significantly contribute to this technology being routinely available in the near future. In order to strengthen the impact of this trial, I suggest that the investigators clarify some of the following issues. The choice of primary outcome is a matter of debate. Time spent in a large glucose range makes sense, as this feature has been extensively used in previous trials involving closed-loop insulin delivery technologies. Indeed, this indicator takes into account the caveats of HbA1c (i.e., glycation gaps), glucose variability, hypoglycemia. However the rationale for the chosen range (70-180 mg/dL) relies upon expert consensus but has not been truly validated by long-term prospective studies. Similarly it is tempting to hypothesize that the increase of time spent in a given range will be beneficial, but correlating time with a clinical outcome is still
--

	speculative. On the other hand, most international therapeutic recommendations are based on HbA1c, which levels have been correlated with the risk of chronic complications of diabetes. Another aspect that is crucial for assessing this new technology is to combine the evaluation of its efficacy and its safety. Whereas time spent in the 70-180 mg/dl may be admitted as a surrogate for the former, time spent below the 70 mg/dl threshold can be a good indicator for the latter. This is especially important to consider for patients that will be included with the lower quartile of HbA1c. Therefore an appropriate primary outcome could be a hierarchised criterium, assessing in a chronological order the time spent in target then time spent in hypoglycemia. May I suggest the investigators to justify the choice of their primary outcome more extensively ? My second concern is about the control group. The strength of this trial is to combine a closed-loop insulin delivery with a pump suspend feature, which increases the overall safety of the device. The control group will use a sole sensor-augmented pump therapy, without any pump suspend feature. Therefore the impact of the closed-loop feature by itself will not be assessed appropriately. The use of pump suspend features (either Low Glucose Threshold Suspend System, or Low Glucose Predictive Suspend system) has now been extensively studied in adult and youth populations of T1D patients, and the efficiency of these systems on the reduction of time spent in hypoglycemia is established. Therefore it is likely that these systems will overcome plain sensor-augmented pump therapy. Whether one or another of these two suspend systems should have been chosen in the comparator group, both from an ethical and methodological point of view, can be questioned. In addition, we also wish that the manuscript clarifies which of the two available pump suspend features (LGTS or LGPS) will be used. Regarding the population, I would appreciate to read about the rationale for discriminating youths and adults with a threshold of 21 years-old. Finally, in order to gather as much information as necessary for filing to health technology assessment agencies, such a large trial could include more data regarding cost-effectiveness assessments (health resources utilisation, real duration of closed-loop system function, duration of visits and hospitalisation etc). Overall I am very supportive of this trial proceeding, which will significantly contribute to this important field of diabetes research.
--	--

REVIEWER	Adam Nicholls University Hospital Southampton
REVIEW RETURNED	28-Mar-2017

GENERAL COMMENTS	Clearly described and justified research protocol suitable for publication.
---

REVIEWER	Jennifer Sherr Yale University New Haven, CT, USA Consultant for Medtronic Diabetes. Advisory board for Bigfoot
-----------------	--

	Biomedical and Insulet Corporation.
REVIEW RETURNED	29-Mar-2017

GENERAL COMMENTS	Summary Bally and colleagues provide the reader with the framework for an upcoming study that the authors are currently conducting. This is a multicenter open-label, randomized controlled trial that seeks to assess the efficacy, safety, and acceptability of closed loop control with the added feature of a pump suspend to sensor augmented pump therapy. The study seeks to enroll 84 participants and the treatment modality will be used for the 3-months in their free-living home setting. The primary outcome will be time in target range (70-180mg/dL) based on sensor glucose values from the 12-week free living phase. A number of secondary outcome measures are specified. Additionally, a subset of participants in the closed loop arm will also invited to take part in some qualitative data collection including use of in-depth interviews and validated questionnaires. The protocol has been approved by the REC and IRB in the UK and ISA and has also been reviewed by regulatory authorities in the UK and the US. The enrollment and recruitment phase of the study commenced in June 2016 with plans for the study to be completed by the end of 2017. Strengths: The premise and protocol for this study are excellent. Study design, outcome measures, and statistical analysis plans are well described. Phenomenal group of investigators who will be working on this project. The results of this study will be of interest to a wide array of individuals. Points for Clarification:  • It seems that one of the novel aspects of the present study is that should closed loop insulin delivery be interrupted for any reason, if sensor glucose values remain available, the participant would be exited to their usual pump settings with the added feature of having the low glucose suspend feature active. This is mentioned in both the background (page 5: lines 50-53), methods and analysis: overview (page 6 lines 7-11), and methods and analysis: safety precautions during CL (page 10 lines 26-30). However, it may be prudent to highlight how important it is. For example, explain that other systems, should closed loop insulin delivery become interrupted, revert to usual basal rates that are pre-programmed in the pump. Potentially, it would be good to highlight that should CL insulin delivery be interrupted when sensor glucose is already low, it would be prudent to have insulin delivery interrupted by having the pump suspend feature activated. • It was unclear if the pump suspend feature is solely for the low glucose suspend feature or if it also includes the predictive low glucose suspend. Minor comments:  • Please note on page 5 on line 21, the sentence should read HbA1c with regular "use" of the device. • Page 16 on line 18, it seems screening and recruitment therefore consider changing the line to "commenced" from "will commence."
---

REVIEWER	J. Herman AMC The Netherlands
REVIEW RETURNED	31-Mar-2017

GENERAL COMMENTS	This is an interesting study design, I do however have some questions/comments:  - Why did the authors chose time in target range as primary outcome variable ? Why not HbA1c (and power the study based on HbA1c)? In addition, how will they handle data from patients not using the device often? This may introduce bias because they may be more likely to be outside of the target range when not using the device often. HbA1c might be a more "objective" measure with regard to the intention to treat analysis. - How did the authors arrive at the exclusion criterium of total daily insulin dose >2 and <15 IU/day? - Is there not the possibility of bias in training time needed (ie more) in the intervention group? If so, how will they handle this? -Page 10, line 55 withdrawel of patients, point 12 (pump suspend): please explain, not really clear -Page 12, line 32: why the cut-off of 16.7 mmol/l?
---

VERSION 1 – AUTHOR RESPONSE

Comments REVIEWER 1	Author Response	Associated Page / Paragraph
The primary objective of this multicenter and international trial is to determine the effectiveness of a closed-loop glucose control system in youths and adults with sub-optimally controlled type 1 diabetes. The strength of the study is to propose a randomised, parallel design trial, as opposed to traditional crossover designs that have been reported so far in this field. The duration of this home study (3 months) will yield clinically relevant informations. The metabolic indicators are clearly informative, and will include HbA1c. A psychosocial evaluation is included and should clarify the question of the acceptability of such a device. The feasibility and overall relevance of this study do not raise any doubt. Overall this trial is very important for the field and should significantly contribute to this technology	We thank the Reviewer for the positive comments.	

being routinely available in the near future. In order to strengthen the impact of this trial, I suggest that the investigators clarify some of the following issues.		
The choice of primary outcome is a matter of debate. Time spent in a target glucose range makes sense, as this feature has been extensively used in previous trials involving closed-loop insulin delivery technologies. Indeed, this indicator takes into account the caveats of HbA1c (i.e., glycation gaps), glucose variability, hypoglycemia. However the rationale for the chosen range (70-180 mg/dL) relies upon expert consensus but has not been truly validated by long-term prospective studies. Similarly it is tempting to hypothesize that the increase of time spent in a given range will be beneficial, but correlating time with a clinical outcome is still speculative. On the other hand, most international therapeutic recommendations are based on HbA1c, which levels have been correlated with the risk of chronic complications of diabetes. Another aspect that is crucial for assessing this new technology is to combine the evaluation of its efficacy and its safety. Whereas time spent in the 70-180 mg/dl may be admitted as a surrogate for the former, time spent below the 70 mg/dl threshold can be a good indicator for the latter. This is especially important to consider for patients that will be included with the lower quartile of HbA1c. Therefore an appropriate primary outcome could be a hierarchised criterium, assessing in a chronological order the time spent in target then time spent in hypoglycemia. May I suggest	We thank the Reviewer for raising this important point. It is generally accepted that both A1c and time in target are important outcomes for closed-loop studies and justifiable as primary outcomes. We agree with the Reviewer that historically, due its well-known and validated link to the development of diabetes complications, HbA1c appears to be an obvious choice for a primary outcome, especially in long-term studies assessing clinical outcomes of a therapeutic intervention. However, as alluded to by the Reviewer, there are certain drawbacks to using HbA1c as the primary outcome measure, specifically the inability to measure and quantify glycaemic excursions. We agree with the Reviewer's comment that correlation between time in target and clinical outcomes has not been validated by long-term prospective studies and its association remains speculative although there is growing support for the use of sensor based outcomes in clinical studies. When designing the study, we were considering possible primary endpoints and it was not an easy decision to come to. Finally the consensus by Principal Investigators following intensive discussions was to adopt time in target as the primary outcome for this specific study. As neither one was considered superior to the other, the decision was thus dependent mainly on the power calculation need of the study, which was based on the cumulative data gained from previous closed-loop studies using time in range as the primary endpoint. In addition,	

the investigators to justify the choice of their primary outcome more extensively ?	the Principal Investigators at the time considered HbA1c more appropriate for longer term studies (> 3 months duration), as the initial use of glucose sensor in this previously sensor-naïve patients may inadvertently/potentially influence HbA1c levels. We agree with the Reviewer that HbA1c could also be a suitable primary endpoint, and time in target as a co-primary outcome to strengthen evidence for the treatment effect. However, the resulting decrease of power would have required an increased sample size, which would not have been feasible due to limited resources available.	
My second concern is about the control group. The strength of this trial is to combine a closed-loop insulin delivery with a pump suspend feature, which increases the overall safety of the device. The control group will use a sole sensor-augmented pump therapy, without any pump suspend feature. Therefore the impact of the closed-loop feature by itself will not be assessed appropriately. The use of pump suspend features (either Low Glucose Threshold Suspend System, or Low Glucose Predictive Suspend system) has now been extensively studied in adult and youth populations of T1D patients, and the efficiency of these systems on the reduction of time spent in hypoglycemia is established. Therefore it is likely that these systems will overcome plain sensor-augmented pump therapy. Whether one or another of these two suspend systems should have been chosen in the comparator group, both from an ethical and	We thank the Reviewer for the insightful comment. From an ethical standpoint, we believe that this issue has been mitigated as the participants recruited to this study are sensor-naïve and have not used pump suspend features previously. The control arm chosen for this study therefore closely reflect their usual care. Regarding the methodological aspect of the control group, when the study was designed the goal was to compare the algorithmic versus non-algorithmic insulin delivery in previously sensor naïve population (regular use of real time in preceding 3 months is an exclusion criteria). Most closed-loop studies to date have adopted the same approach to ours, with the exception of one study by Ly et al (Diabetes Care. 2015 Jul;38(7):1205-11). We agree with the Reviewer that given the increasing evidence of the efficacy of the pump suspension features and their increasing availability future closed-loop studies may review the use of PLGS/LGS as the control arm. We have therefore acknowledge this as a potential limitation of the study on page 3.	Page 5, Para 1 Page 8, Para 1

methodological point of view, can be questioned. In addition, we also wish that the manuscript clarifies which of the two available pump suspend features (LGTS or LGPS) will be used.	We have provided clarification that low glucose suspend (and not predictive low glucose suspend) is used in during the study intervention on page 5.	
Regarding the population, I would appreciate to read about the rationale for discriminating youths and adults with a threshold of 21 years-old.	We thank the Reviewer for the comment. The age threshold was chosen to comply and be in line with the age of minors in the US (which is 21 years, compared to 18 years in the UK). We have therefore clarified this accordingly.	Page 5, Para 1
Finally, in order to gather as much information as necessary for filing to health technology assessment agencies, such a large trial could include more data regarding cost-effectiveness assessments (health resources utilisation, real duration of closed-loop system function, duration of visits and hospitalisation etc).	We thank the Reviewer for the comment. We agree that cost-effective assessments are important to support health technology approval by regulatory agencies and reimbursement policies. Unfortunately, the assigned grant did not provide adequate resources to fulfil such an undertaking appropriately for the current study. We have a study planned which includes health economic evaluation (Clinicaltrials.gov NCT02925299). Data from this upcoming study will support reimbursement decision-making.	
Comments REVIEWER 2	Author Response	Associated Page / Paragraph
Clearly described and justified research protocol suitable for publication.	We thank the Reviewer for the kind commendation.	
Comments REVIEWER 3	Author Response	Associated Page / Paragraph
Bally and colleagues provide the reader with the framework for an upcoming study that the authors are currently conducting. This is a multicenter open-label, randomized controlled trial that seeks to assess the efficacy, safety, and acceptability of closed loop control with the added feature of a pump suspend to sensor augmented pump therapy. The study seeks to enroll 84 participants and the treatment	We thank the Reviewer for the kind commendation.	

modality will be used for the 3-months in their free-living home setting. The primary outcome will be time in target range (70-180mg/dL) based on sensor glucose values from the 12-week free living phase. A number of secondary outcome measures are specified. Additionally, a subset of participants in the closed loop arm will also invited to take part in some qualitative data collection including use of in-depth interviews and validated questionnaires. The protocol has been approved by the REC and IRB in the UK and ISA and has also been reviewed by regulatory authorities in the UK and the US. The enrollment and recruitment phase of the study commenced in June 2016 with plans for the study to be completed by the end of 2017. The premise and protocol for this study are excellent. Study design, outcome measures, and statistical analysis plans are well described. Phenomenal group of investigators who will be working on this project. The results of this study will be of interest to a wide array of individuals.		
It seems that one of the novel aspects of the present study is that should closed loop insulin delivery be interrupted for any reason, if sensor glucose values remain available, the participant would be exited to their usual pump settings with the added feature of having the low glucose suspend feature active. This is mentioned in both the background (page 5: lines 50-53), methods and analysis: overview (page 6 lines 7-11), and methods and analysis: safety precautions during CL (page 10 lines 26-30). However, it may be prudent to highlight how important it is. For example, explain that other systems, should closed loop insulin delivery become interrupted, revert to usual basal	Thank you for the kind comments. As per the Reviewer's suggestion we have highlighted the importance of the pump suspension feature if closed-loop becomes unavailable to avoid hypoglycaemia. We have provided further clarification on the use of low glucose suspend, rather than predictive low glucose suspend feature, during the study intervention.	Page 4, Para 5 Page 5, Para 1 Page 8, Para 1

rates that are pre-programmed in the pump. Potentially, it would be good to highlight that should CL insulin delivery be interrupted when sensor glucose is already low, it would be prudent to have insulin delivery interrupted by having the pump suspend feature activated.  • It was unclear if the pump suspend feature is solely for the low glucose suspend feature or if it also includes the predictive low glucose suspend. 		
Please note on page 5 on line 21, the sentence should read HbA1c with regular “use” of the device.		Page 4, Para 2
Page 16 on line 18, it seems screening and recruitment therefore consider changing the line to “commenced” from “will commence.”	Thank you, we have changed the line as suggested.	Page 15, Para 1
Comments REVIEWER 4	Author Response	Associated Page / Paragraph
This is an interesting study design, I do however have some questions/comments:		
Why did the authors chose time in target range as primary outcome variable ? Why not HbA1c (and power the study based on HbA1c)? In addition, how will they handle data from patients not using the device often? This may introduce bias because they may be more likely to be outside of the target range when not using the device often. HbA1c might be a more "objective" measure with regard to the intention to treat analysis.	We thank the Reviewer for the comment. As per our response to Reviewer 1, we have provided justification for the chosen primary outcome, namely the evaluation of efficacy and safety rather than clinical outcome per se, the use of continuous glucose monitoring as an integral component of the intervention being studied (closed-loop insulin delivery system), and that assessment of time in target range would provide further insight into glycaemic excursions which are not apparent with HbA1c alone, as alluded to by Reviewer 1. In addition, the cumulative data and historical information gained from previous closed-loop studies allows the study to be robustly powered. Thus, we are of the opinion that time in target is more reflective	

	of the objective of the present study which is improvement of glucose control whilst reducing hyper- and hypoglycaemia, which would otherwise be missed by measurement of HbA1c. The analyses will be performed on intention to treat principle, and therefore be independent of closed-loop use during the study and remove the concern of bias in data handling/analysis. HbA1c will be a secondary endpoint to support finding using sensor based endpoints. We acknowledge that unequal sensor glucose availability between the two groups may be a potential confounder of CGM-based glucose outcome data, however clinically significant differences in sensor glucose availability between closed-loop and control group have not been observed in our previous closed-loop studies of similar duration.	
How did the authors arrive at the exclusion criterium of total daily insulin dose >2 and <15 IU/day?	The criteria of minimum and maximum total daily insulin dose were chosen based on qualitative assessments. The former was related to the accuracy of insulin delivery while the latter was considered a sign of potential significant insulin resistance, and was thus chosen as an exclusion criterion. These parameters were chosen to mitigate safety concerns related to closed-loop controller operation (insulin under- and over-delivery) and technical limitations (i.e. insulin pump reservoir volume and minimum infusion rate). The chosen criteria have been reviewed and approved by the relevant regulatory authorities (UK Medicines and Healthcare products Regulatory Agency, and US Food and Drug Administration).	

Page 10, line 55 withdrawel of patients, point 12 (pump suspend): please explain, not really clear	We thank the Reviewer for the comment. The study was designed to contrast algorithmic (closed-loop combined with pump suspension) versus non-algorithmic (sensor augmented pump therapy) insulin delivery. As protocol adherence is key, the decision was made during the study design phase by the Trial Steering Committee that the activation of the suspend feature during the Control period would be deemed a protocol violation. Thus participants who continue to activate the pump suspend feature during the Control period, in spite of being instructed and advised not to, will be withdrawn from the study. This is specified on page 8 “Participants will be instructed not to activate threshold suspend (low glucose or predictive low glucose suspend features).” To date, no such withdrawal occurred.	Page 8, Para 3
Page 12, line 32: why the cut-off of 16.7 mmol/l?	Thank you for the question. We have chosen 16.7mmol/l (300mg/dl), as it refers to the currently accepted definition of significant hyperglycaemia by professional diabetes organisations such as the American Diabetes Association, and more recently the consensus report by artificial pancreas researchers (Maahs et al, Outcome Measures for Artificial Pancreas Clinical Trials: A Consensus Report, Diabetes Care 2016;39:1175–1179 DOI: 10.2337/dc15-2716). The assessment of “significant hyperglycaemia” was also requested by the FDA which approved the study.	

VERSION 2 – REVIEW

REVIEWER	Pierre Y. Benhamou University of Grenoble Alpes, France member of the scientific board of the french Artificial Pancreas DiabeLoop consortium
REVIEW RETURNED	28-Apr-2017

GENERAL COMMENTS	The different points raised by all reviewers have been appropriately addressed in this revised version that is suitable for publication.
--

REVIEWER	Jennifer Sherr Yale University School of Medicine New Haven, CT, USA Consultant for Medtronic Diabetes, Advisory Board for Insulet and Bigfoot Biomedical.
REVIEW RETURNED	27-Apr-2017

GENERAL COMMENTS	Thank you for your responses. All questions and concerns were adequately addressed.
---